# Memorisation Cartography: Mapping out the Memorisation-Generalisation Continuum in Neural Machine Translation

**Verna Dankers**[⋈,*] and **Ivan Titov**[⋈,◊] and **Dieuwke Hupkes**[∞]

[⋈]ILCC, University of Edinburgh
[◊]ILLC, University of Amsterdam
[∞]FAIR, Meta AI
vernadankers@gmail.com, ititov@inf.ed.ac.uk,
dieuwkehupkes@meta.com

## Abstract

When training a neural network, it will quickly memorise some source-target mappings from your dataset but never learn some others. Yet, memorisation is not easily expressed as a binary feature that is good or bad: individual datapoints lie on a memorisation-generalisation continuum. What determines a datapoint's position on that spectrum, and how does that spectrum influence neural models' performance? We address these two questions for neural machine translation (NMT) models. We use the counterfactual memorisation metric to (1) build a resource that places 5M NMT datapoints on a memorisation-generalisation map, (2) illustrate how the datapoints' surface-level characteristics and a models' per-datum training signals are predictive of memorisation in NMT, (3) and describe the influence that subsets of that map have on NMT systems' performance.[1]

## 1 Introduction

When training neural networks, we aim for them to learn a generic input-output mapping, that does not overfit too much on the examples in the training set and provides correct outputs for new inputs. In other words, we expect models to *generalise* without *memorising* too much. Yet, adequately fitting a training dataset that contains natural language data inevitably means that models will have to memorise the idiosyncracies of that data (Feldman, 2020; Feldman and Zhang, 2020; Kharitonov et al., 2021). The resulting memorisation patterns are both concerning and elusive: both memorisation and task performance increase with model size

---

*Work partially conducted during an internship at FAIR.

[1]Click here to interactively explore the NMT memorisation maps in our demo.

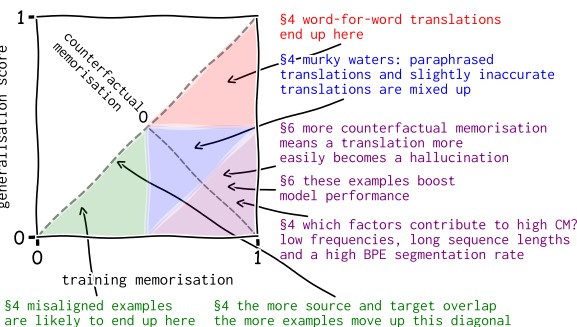

Figure 1: Illustrative summary of findings for different areas of the memorisation map. Counterfactual memorisation subtracts the $y$-coordinate from the $x$-coordinate.

(Carlini et al., 2022); ChatGPT can recall detailed information from its training data, such as named entities from books (Chang et al., 2023); GPT-2 memorises personally identifiable information, including phone numbers (Carlini et al., 2021); GPT-2 and Transformer NMT systems fail to memorise certain idioms (Dankers et al., 2022; Haviv et al., 2023); and in the presence of some source prefixes, NMT systems have memorised always to emit the same translation (Raunak and Menezes, 2022).

These examples illustrate the multi-faceted relation between memorisation and generalisation. In machine learning, memorisation has traditionally been associated with overfitting and overparameterisation; Dietterich (1995, p.326) already discusses the concern of "fit[ting] the noise in the data by memorizing various peculiarities". Yet, for *deep* learning, overfitting has even been referred to as *benign* when models overfit the training data but still have a low generalisation error (Zhang et al., 2017; Bartlett et al., 2020). Besides, memorisation is not just considered detrimental; for instance, it is needed if the natural data distribution is long-tailed

(Feldman, 2020) or if factual information needs to be stored (Haviv et al., 2023). At the same time, benign overfitting is still concerning if it makes models less robust (Sanyal et al., 2021) and introduces privacy concerns (Carlini et al., 2019).

In this work, we get a step closer to understanding the relation between memorisation and datapoints' characteristics for neural machine translation (NMT). Instead of focusing on cases that are memorised verbatim, we take 5M examples from five language pairs, put them on a memorisation-generalisation map, learn to predict examples' places on the map and analyse the relation between the map and model performance. The map is centred around the **counterfactual memorisation** (CM) metric (Zheng and Jiang, 2022) (see §3), and available to readers via our repository. Using the map, we address the following research questions, for which we illustrate takeaways and interesting findings in Figure 1:

1. *How do characteristics of datapoints relate to their position on the memorisation-generalisation map?* In §4, we compute 28 quantitative features and annotate a data subset manually using 7 additional features. We discuss how source-target similarity, input and output length, token frequency and tokens' segmentation relate to the memorisation map.

2. *Can we approximate memorisation metrics using datapoints' characteristics?* In §5, we use datapoints' characteristics to predict memorisation values using multi-layer perceptrons (MLPs) to consolidate findings from §4, to compare different languages to one another and to understand whether resource-intensive memorisation computation has cheaper approximates. We find that the MLPs generalise cross-lingually: characteristics' relation to memorisation is largely language-independent.

3. *How does training on examples from different regions of the memorisation map change models' performance?* Finally, we relate different parts of the map to the quality of NMT systems in terms of BLEU, targets' log-probability and hallucination tendency (see §6). Our results confirm previous work from other tasks – examples with high CM are most relevant for models' performance – yet there are caveats worth mentioning, in particular for the hallucination tendency.

## 2  Related work

How has memorisation been measured in NLP, and what have we learnt as a result? In this section, we first dive into memorisation metrics from NLP generically and then report on the limited set of related work that exists for NMT.

**Memorisation metrics in NLP**  In NLP, memorisation has most often been quantified via binary metrics that identify examples that are memorised *verbatim*. Carlini et al. (2021) measure $k$-**eidetic memorisation** (i.e. a string appears at most $k$ times *and* can be extracted from the model). Other studies omit the constraints on $k$ and simply examine whether, after feeding the context, parts of a sentence can be extracted verbatim (Carlini et al., 2019; Kharitonov et al., 2021; Carlini et al., 2022; Mireshghallah et al., 2022; Tirumala et al., 2022; Chang et al., 2023). Sometimes, the training data is known or even modified to include 'canaries' (e.g. Mireshghallah et al., 2022) while in other cases, the training data is unknown (e.g. for ChatGPT, Chang et al., 2023). These studies have raised privacy concerns – by pointing out that personally identifiable information and copyright-protected text is memorised – and have identified potential causes of memorisation, such as repetition, large vocabulary sizes, large model sizes and particular fine-tuning techniques.

A second approach has been to rate memorisation on a scale (Zhang et al., 2021; Zheng and Jiang, 2022). Zhang et al. measure **counterfactual memorisation** (CM), a metric from computer vision (Feldman and Zhang, 2020) that assigns high values to examples a model can only predict correctly if they are in the training set. Zhang et al. identify sources of CM in smaller language models (trained with 2M examples), such as the presence of non-English tokens in English sentences. Inspired by the CM metric, Zheng and Jiang (2022) use **self-influence** to quantify the change in parameters when down-weighting a training example, to measure memorisation in sentiment analysis, natural language inference and question answering.

**Memorisation metrics in NMT**  For NMT, memorisation is less well explored, but we still observe a similar divide of metrics. Raunak and Menezes (2022) propose **extractive memorisation**, a binary metric that identifies source sentences with a prefix for which models generate the same translation, independent of the prefix's ending. Raunak et al.

([2021](#)) compute CM scores in a low-resource NMT setup to show that hallucinations are more prominent among examples with higher CM values.

We, too, treat memorisation as a graded phenomenon by using CM-based metrics. Whereas Raunak et al. (2021) solely explore CM in the context of hallucinations, we build a multilingual resource of memorisation metrics, examine the characteristics of datapoints that influence their position on the memorisation map, and investigate the relation to models' performance.

## 3 Experimental setup

This section details the memorisation metrics employed and the experimental setup for the model training that is required to compute those metrics.

**Memorisation metrics**  To obtain a graded notion of memorisation, we employ the **counterfactual memorisation** (CM) metric of Feldman and Zhang (2020) and Zhang et al. (2021).[2] Assume an example with input $x$ and target $y$, and a model with the parameters $\theta^{\text{tr}}$ trained on all training data, and $\theta^{\text{tst}}$ trained on all examples except $(x, y)$. CM can be computed as follows:

$$\text{CM}(x, y) = \underbrace{p_{\theta^{\text{tr}}}(y|x)}_{\text{training memorisation}} - \underbrace{p_{\theta^{\text{tst}}}(y|x)}_{\text{generalisation score}}$$

As leaving out individual datapoints is too expensive, computationally, we leave out data subsets, similar to Zhang et al. (2021). To then compute the CM of an individual datapoint, one can collect models and average the target probability $p_{\theta_n^m}(y|x)$ over all models $\theta_n^m \in \Theta^m$ in a collection, where $\Theta^{\text{tr}}$ and $\Theta^{\text{tst}}$ represent sets of models that did and did not train on $(x, y)$, respectively. Since we consider the generation of sequences, we also aggregate probabilities over tokens in the target sequence of length $\ell$ using the geometric mean. We combine averaging over models and averaging over tokens in a likelihood metric (LL), that is computed for $\Theta^{\text{tr}}$ and $\Theta^{\text{tst}}$ to obtain the CM value of an example:

$$\text{LL}(x, y, \Theta^m) = \frac{1}{|\Theta^m|} \sum_{n=1}^{|\Theta^m|} \left( \prod_{t=1}^{\ell} p_{\theta_n^m}(y_t|y_{<t}, x) \right)^{\frac{1}{\ell}}$$

In Appendix D.3, we replace the probability-based measure with BLEU scores for greedily decoded

---

[2]We refer to it as *counterfactual memorisation*, following work of Zhang et al. (2021). The metric is also known as *label memorisation* in other articles.

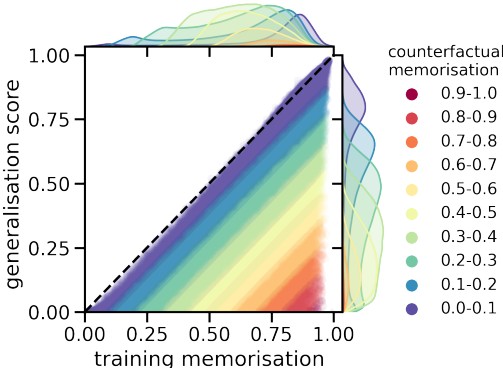

Figure 2: The memorisation map for EN-ES. Colours indicate counterfactual memorisation from 0 (dark blue, along the diagonal) to 1 (dark red, bottom right).

hypotheses and reproduce a subset of the findings with those alternative maps.

CM thus consists of two components which we refer to as **training memorisation** (TM, that expresses how well a model performs on training examples) and the **generalisation score** (GS, that expresses how well a model performs on unseen examples). CM – the difference between the two – is thus high for examples that can be predicted correctly if they are in the training set but that a model cannot generalise to if they are not. Instead of approaching CM as a one-dimensional metric, we examine patterns that underlie all three metrics.

**Data**  Even when leaving out data subsets, computing the memorisation metrics is still resource-intensive. To balance the efficiency of the computation with the quality of the NMT systems, we use corpora with 1M examples for five language pairs: English is the source language, and the target languages are German, Dutch, French, Spanish, and Italian. To enable direct comparison between languages, we collect parallel data by taking sentence pairs from the intersection of the OPUS corpora for these languages (Tiedemann and Thottingal, 2020). Using multiple languages aids in assuring that the conclusions are not language-specific. Appendix A.1 details how the corpora were filtered and the 1M examples were selected. The resulting data is relatively 'clean'. What happens when using a *random* OPUS subset with much more noisy data? Appendix D.1 elaborates on this.

**Training models to obtain memorisation metrics**  We train 40 models to compute our metrics repeatedly on a randomly sampled 50% of the training data, while testing on the remaining 50%. The mod-

els are `transformer-base` models (Vaswani et al., 2017), trained with `fairseq` for 100 epochs (Ott et al., 2019). To ensure that this leads to reliable CM scores, we compare the scores computed over the first 20 seeds to those computed using the second 20 seeds: these scores correlate with Pearson's $r=0.94$. When combining 40 seeds, the metrics are thus even more reliable. We evaluate our models using the FLORES-200 'dev' set (Costa-jussà et al., 2022), a dataset created by expert translators for Wikimedia data. Appendix A.2 provides details on model training and the development set's BLEU scores.

## 4 Data characterisation: what lies where on the memorisation map?

We now have values for our memorisation metrics for 5M source-target pairs across five language pairs. We can view each source-target pair as a coordinate on a map based on the train and test performance associated with that example; the offset of the diagonal indicates the CM. Figure 2 illustrates the coordinate system for EN-ES. It represents datapoints using scattered dots, coloured according to CM. As is to be expected, the TM values exceed the GS values, meaning that generating an input's translation is easiest when that example is in the training set. Examples with high CM are rare: few examples are *very* easily memorised during training while also having a *very* low generalisation score. Our interactive demo can be used to examine individual instances on the map.

To better understand which characteristics influence a datapoint's position on this map, we next analyse the correlation between datapoints' features (automatically computed and manually annotated ones) and different regions of this landscape.

### 4.1 Analysis of feature groups

We compute 28 language-independent features that together we believe to cover a broad spectrum of surface-level features from both the source and target. 19 features describe the source and target separately based on the length, word frequency, the number of target repetitions, BPE segmentation, and digit and punctuation ratios. The nine remaining features capture the source-target overlap with the edit distance, edit distance of the target's back translation (computed with models from Tiedemann and Thottingal, 2020), length differences, ratios of unaligned words, word/token overlap and the alignment monotonicity (as per the

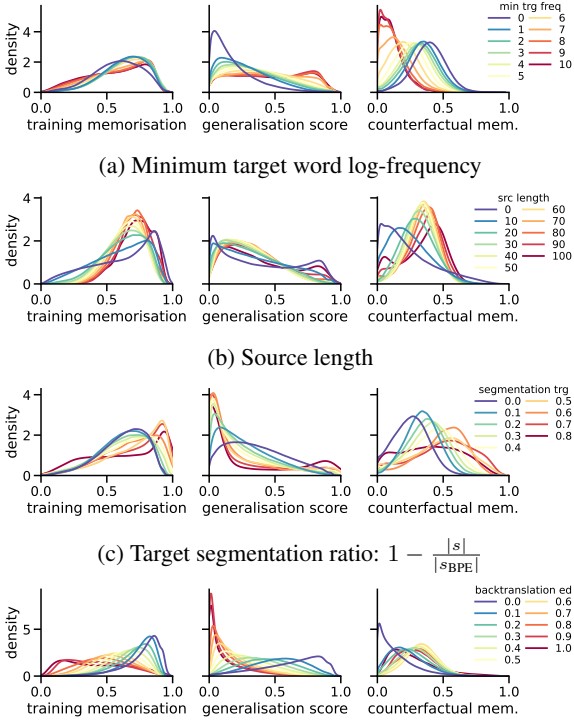

(a) Minimum target word log-frequency

(b) Source length

(c) Target segmentation ratio: $1 - \frac{|s|}{|s_{\mathrm{BPE}}|}$

(d) Backtranslation edit distance between source and target

Figure 3: Illustration of how four different features relate to the three memorisation metrics. Appendix C displays these graphs for additional features.

Fuzzy Reordering Score, Talbot et al., 2011). For each feature, we compute Spearman's rank correlation ($\rho$) for TM, GS and CM, combining datapoints from all five language pairs. All correlations are contained in Figure 14, and we report the most prominent patterns in the remainder of this subsection. Appendix C provides implementation details per feature.

**Frequency** The frequency features are strong predictors for CM (e.g. for the minimum target log-frequency feature, $\rho_{\mathrm{CM}}=-0.46$, depicted in Figure 3a). Examples with low-frequency tokens can be learnt during training, but models are much less likely to assign a high probability to targets with low-frequency tokens during testing.

**Length** The length characteristics correlate more strongly with CM than with TM or GS (e.g. for the source length, $\rho_{\mathrm{CM}}=0.26$, also visualised in Figure 3b). This means that longer sequences tend to have a larger difference in performance between training and testing time, compared to shorter sequences.

**Token segmentation**    Thirdly, the segmentation of tokens into subtokens positively correlates with CM ($\rho_{CM}$=0.40/$\rho_{CM}$=0.37 for source/target segmentation, respectively), as is shown in Figure 3c. The segmentation compares the number of whitespace-based tokens to BPE tokens: $1 - \frac{|s|}{|s_{BPE}|}$.

**Repetitions**    A feature that is a positive predictor for TM and GS is the repetition of the target ($\rho_{TM}$=0.15, $\rho_{GS}$=0.22, $\rho_{CM}$=−0.15). This is expected, considering that similar targets have similar sources and are thus more easily memorised. Previous work already noted that repetition-related characteristics (repeated sentence 'templates') lead to high TM (Zhang et al., 2021).

**Source-target overlap**    The remaining features that correlate rather strongly with TM and GS are: the target's backtranslation edit distance to the source ($\rho_{TM}$=−0.49, $\rho_{GS}$=−0.56, see Figure 3d), the source-target edit distance ($\rho_{TM}$=−0.30, $\rho_{GS}$=−0.26), and the fraction of unaligned tokens ($\rho_{TM}$=−0.32, $\rho_{GS}$=−0.32 for target tokens, $\rho_{TM}$=−0.28, $\rho_{GS}$=−0.29 for source tokens).[3] Apart from negative correlations, there are weak positive predictors, e.g. token overlap ($\rho_{TM}$=0.15, $\rho_{GS}$=0.13) and digit ratio ($\rho_{TM}$=0.13, $\rho_{GS}$=0.07). These features express (a lack of) source-target overlap: source words are absent in the target, or vice versa. Because they are predictive of both TM and GS, they are not that strongly correlated with CM: they predict where along the diagonal an example lies but not its offset to the diagonal. While you might expect that examples with little source-target overlap *require* memorisation, their TM values remain low throughout the 100 epochs. The only relation to CM we observe is that, typically, examples in the mid-range (i.e. with some overlap) have higher CM than examples with extreme values (i.e. full or no overlap). CM thus highlights what models can memorise in a reasonable amount of training time. This provides a lesson for NMT practitioners: models are unlikely to memorise the noisiest examples, which might be one of the reasons why semi-automatically scraped corpora, rife with noisy data, have driven the success behind SOTA NMT systems (e.g. Schwenk et al., 2021a,b).

---

[3]Backtranslations are computed using `Marian-MT` models trained on OPUS (Tiedemann and Thottingal, 2020) to ensure the accuracy of the feature. Alignments are computed with `eflomal` (Östling and Tiedemann, 2016).

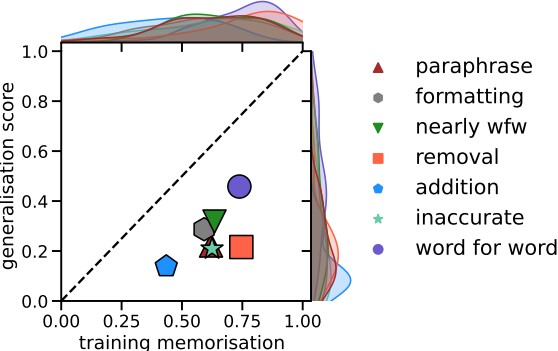

Figure 4: Centroids and marginal distributions of examples grouped through the manual annotation for EN-NL.

## 4.2    Manual annotation

The previous subsection discussed coarse patterns that relate datapoints' features to memorisation metrics. To understand whether similar patterns appear when we qualitatively examine source-target pairs, we annotate 250 EN-NL examples, uniformly sampled from different parts of the coordinate system, with lengths $l$ for which $10 < l < 15$. We annotate them using the following labels: (nearly) a word-for-word translation; paraphrase; target omits content from the source; target adds content; inaccurate translation; target uses different formatting in terms of punctuation or capitalisation. Multiple labels can apply to one example. For an elaboration on how the labels were assigned and processed, see Appendix C. Figure 4 summarises the results. Firstly, these results consolidate the observation regarding source-target overlap: word-for-word translations, e.g. Example (1), are positioned closer to the top right corner compared to inaccuracies, e.g. Example (2), and paraphrases, e.g. Example (3).

(1)  $s$ EN: Leave a few empty rows and columns on either side of the values.
     $t$ NL: Laat enkele rijen en kolommen leeg aan beide zijden van de waarden. (TM=0.85, GS=0.55)

(2)  $s$ EN: The last 2 years of my life has been one big lie.
     $t$ NL: "De afgelopen twee jaren van mijn leven zijn een grote leven geweest. (*leven != lie*, TM=0.28, GS=0.14)

(3)  $s$ EN: I don't know how she did it, but she did it.
     $t$ NL: Geen idee hoe, maar ze deed 't. (*underlined portions are paraphrases*, TM=0.23, GS=0.14)

Yet, paraphrases and inaccurate translations have similar centroids on the map; the differences between those two types are subtle and are not well reflected in the memorisation metrics. Lastly, what is not that easily captured by one automated feature, but does show up in these results, is that targets that remove content from the source, e.g. Example (4), are easier to memorise during training than those that add content, e.g. Example (5).

(4) $s$ EN: Then we had our little adventure up in Alaska and things started to change.
    $t$ NL: Toen waren we in Alaska en begonnen dingen te veranderen. (TM=0.80, GS=0.22)

(5) $s$ EN: There are periods and stages in the collective life of humanity.
    $t$ NL: Evenzo zijn er perioden en fasen in het collectieve leven van de mensheid. (TM=0.45, GS=0.34)

### 4.3 Comparing metrics for five languages

The trends of what complicates or eases memorisation are consistent for all five languages, upon which we elaborate in §5. Does this mean that one source sentence will have a very similar memorisation score across the five different languages in our parallel corpus? Not necessarily, as is shown in Figure 5, which, for the three memorisation metrics, reports the correlation between scores associated with the same source sequence (but different target sequences) across the different languages.

Source sequences with different positions on the memorisation maps from two language pairs give insight into how the relation between the source and target affects memorisation. Examples that move from the top right in one language to the bottom left in another show how targets go from easily learnable to unlearnable: in Examples (6) and (7) target 2 ($t_2$) seems misaligned. In Example (8) $t_2$ is contextually relevant but not a translation.

(6) $s$ EN: She's not a child anymore.
    $t_1$ ES: Ya no es una niña.
    $t_2$ DE: Du hast das Kind verwöhnt, Matthew. (*You spoiled the child, Matthew*)

(7) $s$ EN: It is an international obligation.
    $t_1$ ES: Es una obligación internacional.
    $t_2$ FR: Nianias sur l'opportunité de cet embargo. (*Nianias on the advisability of this embargo*)

(8) $s$ EN: It's a long story.

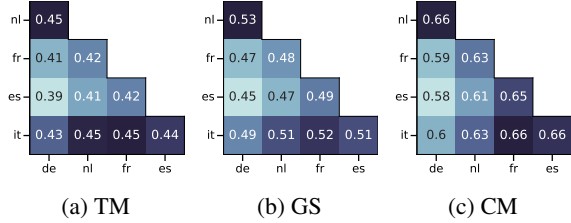

(a) TM    (b) GS    (c) CM

Figure 5: Comparison of the memorisation metrics across the five languages, using Pearson's $r$.

    $t_1$ NL: Het is een lang verhaal.
    $t_2$ IT: Sarebbe troppo lungo spiegarsi. (*It takes too long to explain*)

What about examples that move from the top right to the bottom right, i.e. go from easily learnable to only learnable if they are in the train set? Generally, they seem to deviate from source sequences in more subtle ways – e.g. they are missing a term or phrase, as is the case in Examples (9), (10) and (11).

(9) $s$ EN: Kenneth, what are you doing here?
    $t_1$ ES: Kenneth, ¿qué haces aquí?
    $t_2$ FR: Que fais-tu ici (*What are you doing here?*)

(10) $s$ EN: Is this a hunting game?
    $t_1$ FR: C'est un jeu de chasse?
    $t_2$ NL: Is dit een spelletje? (*Is this a game?*)

(11) $s$ EN: We need a viable suspect.
    $t_1$ ES: Necesitamos un sospechoso viable.
    $t_2$ DE: Wir brauchen einen Verdächtigen. (*We need a suspect*)

## 5 Approximating memorisation measures

Having examined correlations between datapoints' features and memorisation values, we now go one step further and treat this as a regression problem: given the characteristics of a datapoint, can we predict memorisation values? We include the previously mentioned features and additional ones obtained from an NMT system during training. We examine the performance of our feedforward predictors and explore how well the predictors generalise across languages. The analysis aids in consolidating findings from §4 and improves our understanding of how language-independent our findings are. Since computing CM is resource-intensive, the predictors can also serve as memorisation approximators (we circle back to this in §6.2).

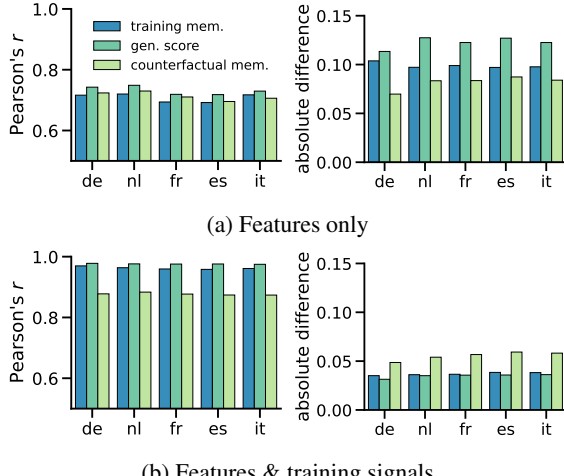

(a) Features only

(b) Features & training signals

Figure 6: Predicting memorisation using an MLP, based on examples' features and models' training signals. The MLP is trained on EN-DE and applied to all languages.

**Experimental setup** To extract training signals, we train one Transformer model per language pair on the full dataset, acting as our *diagnostic run* from which we extract the following signals: 1) the confidence and 2) variability of the target likelihood (Swayamdipta et al., 2020, mean and standard deviation computed over epochs), 3) the target likelihood in the final epoch, 4) a forgetting metric (a counter that accumulates drops in target likelihood during training, Toneva et al., 2019), 5) the hypotheses' likelihood in the final epoch, and 6) metric 1 subtracted from 3, since initial experiments suggested those two correlated most strongly with GS and TM. Next, we train a shallow MLP to predict the memorisation metrics. We train one MLP on the datapoints' features from §4, and one on the features and the training signals, and report their performance using Pearson's correlation and the absolute difference between predictions and memorisation scores. Appendix B details the experimental setup of training the MLPs.

**Results** Here, we show the results of MLPs trained on EN-DE and applied to all other language pairs only. The predictions of the MLP trained only on datapoints' characteristics already positively correlate with the memorisation metrics, with Pearson's $r$ around 0.7 and a mean absolute difference around 0.1, see Figure 6a. Combining the features and training signals further boosts performance (see Figure 6b).

Since we applied the EN-DE MLPs to the other languages, these figures illustrate that an MLP trained on one language is transferrable to mod-

els trained with other target languages. Note that this does not mean that models for the different languages behave similarly for the same source sentences, but instead that models trained on different language pairs behave similarly for source-target pairs with the same features. In practice, this means that we can make an educated guess about the amount of memorisation required for a new datapoint or the same source sequence in another language, using predictors trained on a subset of the data or using a different but related language.[4]

## 6 Memorisation and performance

Finally, we examine the relation that different regions of the map have to models' performance. Firstly (in §6.1), by leaving out data subsets while training using our training examples from §3, and, secondly (in §6.2), by sampling specialised training corpora from a larger dataset of 30M examples. The previous sections showed that results across language pairs are highly comparable, which is why in this section, we employ EN-NL data only.

### 6.1 Importance of different regions

How do examples from specific regions of the memorisation map influence NMT models trained on that data? We now investigate that by training models while systematically removing groups of examples.

**Experimental setup** We train models on datasets from which examples have been removed based on the coordinates from the memorisation map. For 55 coordinates $(i, j)$, where $i, j \in \{.1, .2, \ldots, 1\}$, $j \leq i$, we create training subsets by removing the nearest examples (up to 750k source tokens total). For each training subset, we then train models with three seeds. Depending on the number of examples surrounding a coordinate, the datapoints can lie closer or further away before reaching the limit.

We evaluate models according to three performance metrics: (i) **BLEU** scores for the FLORES-200 dev set (Goyal et al., 2022); (ii) the mean **log-probability** of a target, averaged over datapoints from the FLORES-200 dev set; and (iii) **hallucination tendency** computed using the approach of Lee et al. (2018), which involves the insertion of a token into a source sentence and repeating that for more than 300 tokens (high-frequency, mid-frequency and low-frequency subtokens and punc-

---

[4]We comment on the set of languages used in the limitations section, see §7.

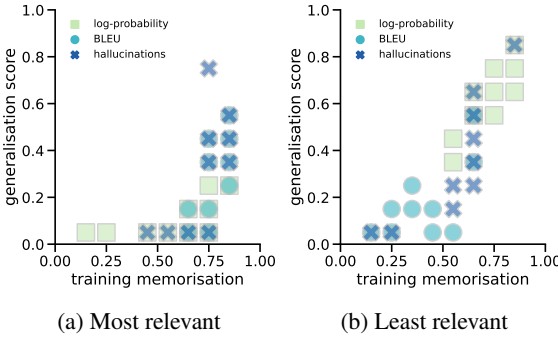

(a) Most relevant      (b) Least relevant

Figure 7: The 10 worst- and 10 best-performing regions on the memorisation map per performance metric.

tuation marks), and four token positions. A hallucination is recorded if BLEU (Lee et al., 2018) drops below 1 after an insertion. We apply this to 1000 examples (500 from FLORES, 500 from our parallel OPUS) and, following Lee et al. (2018), measure the ratio of source sequences which can be perturbed to a hallucination.

**Results** To express an example's impact, we average the performance of all models for which that example was *not* in the training set. The more negatively the performance is affected, the more important an example is. We aggregate over regions of examples and exclude regions that represent <2k datapoints. We then compute the ten most relevant regions (Figure 7a) and the ten least relevant ones (Figure 7b). *Most* relevant means that the BLEU score or log-probability decreases the most if you remove this group or that the hallucination tendency increases the most. *Least* relevant means the opposite. In general, the figures suggest that examples with a higher CM value are more beneficial, and examples closest to the diagonal are the least relevant. This is in accordance with related work from computer vision (Feldman and Zhang, 2020) and NLP classification tasks (Zheng and Jiang, 2022), where examples with high CM values had a larger (positive) contribution to the accuracy than examples with lower CM values.

Why might this be the case? In image classification, Feldman and Zhang (2020) observe that training examples with high CM usually are atypical 'long-tail' examples and mainly improve performance on visually similar test examples. Analogous processes might be at play for translation. Yet, there may be benefits to examples with high CM values even without similar test examples. Preliminary investigations (see Appendix D.2) suggest that there is less redundancy among examples

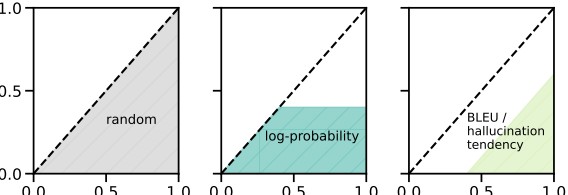

Figure 8: Areas from which we select examples when specialising for a certain metric.

with high CM values (evidenced by more unique $n$-grams), making them more informative training data. Secondly, removing examples with a high CM score negatively affects models' predictions for low-probability tokens, in particular; therefore, including them as training material may quite literally reserve probability mass for the long tail of the output distribution.

### 6.2 Specialising NMT systems using memorisation metrics

In §6.1, we related the maps to models' performance, but all within our original 1M EN-NL datapoints. To understand whether our findings extrapolate to a larger dataset, we perform a *proof-of-concept* study to show that we can put the lessons learnt to use with new data: memorisation metrics can be predicted using datapoints' features and distinct regions of the map have different roles. We now use these lessons for targeted model training.

**Experimental setup** We again train NMT systems in a low-resource setup, yet, different from the previous sections, we now select examples from a larger set of OPUS examples for EN-NL (30M examples) based on their memorisation score as *predicted* using the features-only MLP from §4. We first sample 1M random examples, and then sample one dataset based on the most beneficial region for the log-probability metric, and one dataset based on the region that is most beneficial for BLEU and the hallucination tendency. We mark the regions on the memorisation map in Figure 8. Examples are randomly sampled from those areas until they match the random dataset in the number of tokens. For those three datasets, we train three model seeds.

**Results** We compare the specialised models to a model trained on 1M random examples (Figure 9) and observe that, indeed, the models are somewhat specialised, with the largest relative improvement observed for the hallucination tendency. During training, examples with higher (predicted)

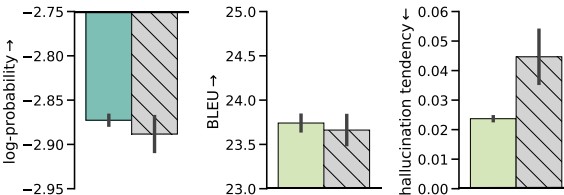

Figure 9: Results of comparing specialised models (in colour) to models trained on randomly selected OPUS data (in gray, with hatches). Error bars show the SE.

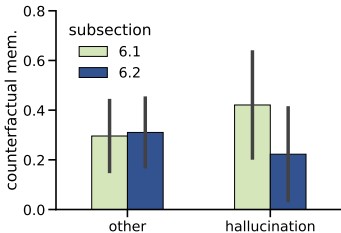

Figure 10: Using all hallucinations from in §6.1 and §6.2, we trace the CM score of the unperturbed sequence and display the distributions. Error bars report SD.

CM scores can thus be beneficial. At the same time, Raunak et al. (2021) reported that when trying to elicit hallucinations from the model using its training examples, examples with high CM scores lead to more hallucinations. To determine whether our results also reflect that, Figure 10 displays the distribution of CM scores associated with each hallucination from the current and previous subsection. For the models from §6.1 (trained on these examples), but not §6.2 (not trained on these examples), hallucinations are indeed more associated with examples with higher CM scores. Together, these results mean that the examples with high CM scores are beneficial training examples, but when generating translations using models trained on them, they are more likely to turn into a hallucination than examples with lower CM scores.

All in all, the small-scale experiment presented here provides a *proof-of-concept*: even when using heuristics (i.e. applying the MLP to new datapoints) we can start to use memorisation metrics in a deliberate way when training NMT systems. However, the hallucination results underscore that the relation between examples with high CM scores and model performance is not straightforward: examples that are most beneficial for systems' quality can introduce vulnerabilities at the same time.

## 7   Conclusion

Learning the input-output mapping that is represented by NMT training data involves so much

more than simply learning a function that translates words from one language into another and rearranges words. It requires understanding which words form a phrase and should be translated together, which words from the source should be ignored, which words can be copied from source to target, and in which contexts "eggs in a basket" are no typical eggs. NMT systems *need* memorisation of patterns that are out of the ordinary.

There are, however, many open questions regarding what memorisation is, when it is desirable and how to measure it. In this work, we took a step toward answering those by creating a map of the memorisation landscape for 5M datapoints. We used graded metrics based on CM to position each example on the memorisation map. We identified salient features for each of the metrics (§4), illustrated that we can approximate memorisation metrics using surface-level features (§5) and drew connections between models' performance and regions of the memorisation map (§6). We found that findings from other tasks and domains about CM transfer to NMT: CM highlights examples that contribute most to models' performance.

Furthermore, our results illustrate that memorisation is not one-dimensional: CM assigns similar scores to paraphrases and slightly inaccurate translations, examples with high CM scores can be beneficial and introduce vulnerabilities at the same time, and there are nuances about which region of the map is most beneficial depending on the performance metric used. We recommend caution when discussing different phenomena under the umbrella term of 'memorisation'. Instead, we encourage future work examining more memorisation maps to further our understanding of the intricacies of task-specific memorisation patterns.

## Limitations

We identify four main limitations with our work:

- The experimental setup used is rather **computationally expensive** due to the repeated model training as explained in §3. We counteracted this by opting for a much, much smaller dataset than state-of-the-art NMT systems would use (OPUS contains hundreds of millions of examples per high-resource language pair), but it still limits the applicability of the methodology to other tasks and for other researchers.

- We did not investigate the impact of major changes to the **experimental setup**, such as using a different model or model size or using a different or a larger dataset. Even though our findings are expected to extend beyond our specific experimental setup, the precise memorisation scores we obtained are specific to our setup – e.g. a larger system is likely to memorise more, and systems trained for much much longer are likely to see increased memorisation. We do experiment with the data used in Appendix D.1.
- We discussed memorisation based on signals that can be observed after model training based on models' outputs. There is the underlying assumption that memorisation happens over time and can thus be observed post-training. However, memorisation not only happens over time, it is also expected to manifest in a particular way in the **space of the model parameters** (e.g. see Bansal et al., 2022), which might not be observable by inspecting output probabilities of tokens. We recommend the analysis of spatial memorisation in NMT as future work.
- All the languages that we consider in this article are considered to be **high-resource languages**. While these might not be the languages most in need of language technology or analysis, the experimental setup of using a parallel corpus limited our possibilities to include lower-resource languages. When taking the intersection of existing NMT corpora, there were not enough remaining examples when including low-resource languages. In preliminary experiments, we also experimented with Afrikaans (together with German and Dutch), and many of the qualitative patterns observed also applied to memorisation measures computed for Afrikaans. If we had used five languages but without the parallel data, it would, however, have been hard to distinguish changes in the metrics due to differences in the dataset from differences between the languages.

## Acknowledgements

We thank Brenden Lake and Jake Russin for their input during the early stages of this project, and thank Max Müller-Eberstein for his comments on the write-up. VD is supported by the UKRI Centre for Doctoral Training in Natural Language Processing, funded by the UKRI (grant EP/S022481/1) and the University of Edinburgh, School of Informatics and School of Philosophy, Psychology & Language Sciences. IT is supported by the Dutch National Science Foundation (NWO Vici VI.C.212.053).

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

# A Experimental setup

## A.1 Data collection

To construct a parallel corpus of 1M sentence pairs across five language pairs, we obtain the `eng-deu`, `eng-fra`, `eng-nld`, `eng-spa` and `eng-ita` data from `https://github.com/Helsinki-NLP/Tatoeba-Challenge/tree/master/data` (version `v2021-08-07.md`). The raw intersection contained 4M sentence pairs, from which we select sentences based on four criteria:

1. The length of the source divided by the length of the target is between $\frac{2}{3}$ and $\frac{3}{2}$.
2. The punctuation ratio of the source and target sequence lies below 0.5.
3. Less than 30% of the words in the source can appear in the target, as well.
4. More than 90% of the numbers contained in the target should also appear in the source sequence.

The Tatoeba repository has the license `Attribution-NonCommercial-ShareAlike 4.0 International`, which allows us to use and redistribute the data, given appropriate attribution.

## A.2 Model training

Before commencing training, we tokenise the data using the Moses tokeniser,[5] and then compute subtokens using *byte pair encodings* (BPE)[6] (Sennrich et al., 2016) to create a joint vocabulary per language pair, with a size of 64k tokens. Model training was performed using Fairseq, version 0.12.1.[7] We train `transformer-base`: 6 encoder and 6 decoder layers, with embedding size and hidden size of 512, and a feedforward size of 2048. All models were trained used the following setup, with the number of total training steps being dependent on the experiment conducted:

- To obtain memorisation scores in §3, we trained for 100 epochs on training datasets of 500k sentence pairs. This involves model training beyond the point of convergence to investigate memorisation.
- The remaining models discussed in the paper are all trained for 50 epochs.

We train using the following command, modelled after exemplar Fairseq translation setups. We did not further tune hyperparameters but did increase `max-tokens` to better utilise the GPU capacity.

```
fairseq-train <DATA_DIR> \
    --arch <MODEL> --save-dir <MODEL_DIR> --share-all-embeddings \
    --fp16 --max-update 200000 \
    --optimizer adam --adam-betas '(0.9,0.98)' --clip-norm 0.0 \
    --lr 0.0005 --lr-scheduler inverse_sqrt \
    --warmup-updates 4000 --warmup-init-lr '1e-07' \
    --label-smoothing 0.1 --criterion label_smoothed_cross_entropy \
    --dropout 0.3 --weight-decay 0.0001 \
    --max-tokens 10000 --update-freq 2 \
    --save-interval 50 --max-epoch <MAXEPOCH> \
    --seed <SEED> --validate-interval 5 \
    --eval-bleu --eval-bleu-args '{"beam":5}' --eval-bleu-remove-bpe
```

Tesla V100-SXM2-32GB GPUs are used for model training in §3. We train each model on a single GPU, on which one epoch of a 500k training set lasted up to 4 minutes, and full training approximately 6 hours. Training all seeds for the five language pairs thus cost 1.2k GPU hours. In §6 we train 3 seeds for 54 coordinates using NVIDIA A100-SXM-80GB GPUs, and the training of one model can take up to 2.5 hours. This thus cost approximately 400 GPU hours.

Figure 11 illustrates the BLEU scores on a development set over the course of training. At the time of writing, the

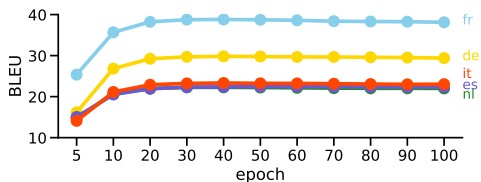

Figure 11: BLEU on the evaluation dataset FLORES-200, when training to obtain the memorisation scores.

top FLORES-200 'dev' performances on the OPUS-MT leaderboard are 40.4, 27.1, 51.5, 28.0 and 29.2 for DE, NL, FR, ES and IT, respectively. Of course, our models trained on a fraction of SOTA MT datasets underperform, but our relative differences in BLEU across languages are similar.

---

[5] `https://github.com/moses-smt/mosesdecoder/blob/master/scripts/tokenizer/tokenizer.perl`
[6] `https://github.com/rsennrich/subword-nmt`
[7] `https://github.com/facebookresearch/fairseq`

# B Extended discussion of approximating memorisation measures from §5

In §5, we questioned whether there is a way to predict memorisation metrics without actually computing counterfactual memorisation and the generalisation score, which are 'expensive' since they can only be computed when examples are *not* present in the training set. To do that, we trained one model per language pair for 50 epochs, acting as our *diagnostic run* from which we estimate memorisation measures using training signals obtained for the full dataset:

- **Confidence** and **variability**: the mean and standard deviation of the target likelihood averaged over all epochs (adapted metrics from Swayamdipta et al., 2020);
- **Final train likelihood**: the likelihood of the target in the final training epoch;
- **Forgetting**: the sum of all decreases in target likelihood observed for consecutive epochs (adapted metric from Toneva et al., 2019);
- **Hypotheses' likelihood** obtained in the final epoch. Uncertainty can aid in detecting out-of-domain data (D'souza et al., 2021), and hallucinations (Guerreiro et al., 2023);
- We also included **final train likelihood - confidence** since initial experiments suggested those two correlated most strongly with training memorisation and generalisation score, and counterfactual memorisation is known to be a combination of those two signals.

Apart from the hypotheses' likelihood, these signals are ones you would naturally obtain while training a model using teacher forcing. Firstly, we inspect to what extent these signals naturally correlate with memorisation measures. Afterwards, we train shallow MLPs to predict the memorisation metrics: firstly only using the datapoints' features discussed in §4, and, secondly, using the features and training signals. Those MLPs were trained for 20 epochs maximum, using Adam according to default hyperparameters in the `sklearn.neural_network.MLPClassifier` class. The MLP takes 28 inputs when training with features only, and 28 + 6 when adding the training signals, and has two hidden layers of 100 units. It predicts all memorisation metrics at the same time.

**Results** Do these signals correlate with memorisation measures? Figure 12a shows this for EN-NL. The strongest correlations (around 0.95 Pearson's $r$) is observed between the confidence and the generalisation score. The confidence score expresses *when* an example is learnt during training, as it will be high for examples that are learnt immediately, low for examples that are learnt very late, and close to zero for examples that are not learnt at all. These results suggest that that temporal indication of when an example is learnt, strongly correlates with the performance a model would have on an example, if that example had been in the test set.

For counterfactual memorisation, correlations are much lower (understandably so since it depends both on train *and* test likelihood), even when we look at the combined feature. As discussed in the main text, we can improve upon this by training an MLP to predict the memorisation measures from both the features and the training signals. The main text discussed this for MLPs trained on EN-DE and applied to other languages, but we can train on any language and still obtain strong results on the other languages. Figure 12b illustrates this for CM.

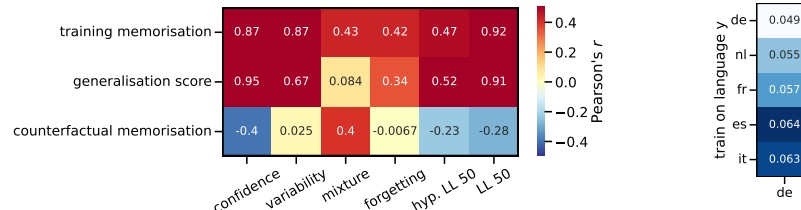 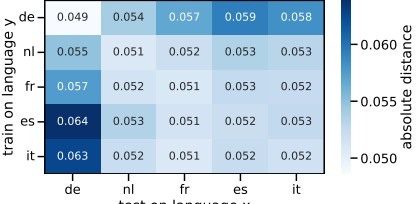

(a) Correlations between training signals and metrics (EN-NL)   (b) Applying MLPs across languages when predicting CM

Figure 12: Correlations for predictive signals and memorisation measures in terms of Pearson's $r$, and comparisons of MLPs' applicability across languages in terms of the absolute difference between the predictions and the target.

## C Extended data characterisation feature analysis from §4

**Experimental setup**

- Counterfactual memorisation is capped at 0 since a negative counterfactual memorisation score is likely to be a side effect of noise in the estimation of the memorisation measures, and typically the train likelihood of the target should outperform the test likelihood. The examples for which we had to cap this measure are rare.

- The features we analyse fall into three categories:
  (1) Source/target-only features

    - $|s|$ source length (x2, BPE tokenised and white space tokenised)
    - $|t|$ target length (x2, BPE tokenised and white space tokenised)
    - $\frac{|s|}{|t|}$ (x2, BPE tokenised and white space tokenised)
    - Average log frequency of source tokens (x2, BPE tokenised and white space tokenised)
    - Average log frequency of target tokens (x2, BPE tokenised and white space tokenised)
    - Minimum log frequency of source tokens (x2, BPE tokenised and white space tokenised)
    - Minimum log frequency of target tokens (x2, BPE tokenised and white space tokenised)
    - Number of repetitions of this target
    - Segmentation of the source: $1 - \frac{|s_{ws}|}{|s_{BPE}|}$, 0 means no segmentation beyond the token level
    - Segmentation of the target: $1 - \frac{|t_{ws}|}{|t_{BPE}|}$, 0 means no segmentation beyond the token level
    - Digit ratio: how many tokens in the source are digits
    - Punctuation ratio: how many tokens in the source are punctuation

  (2) Source-target interaction features

    - Token-level Levenshtein edit-distance between source and target
    - Comparison by backtranslation, obtained with `Marian-MT` models trained on OPUS by Tiedemann and Thottingal (2020), by computing the token-level Levenshtein edit-distance between source and target
    - $|s| - |t|$ (2x, BPE tokenised and white space tokenised)
    - Ratio of unaligned source words, alignments are obtained with `eflomal` (Östling and Tiedemann, 2016)
    - Ratio of unaligned target words, alignments are obtained with `eflomal` (Östling and Tiedemann, 2016)
    - Alignment monotonicity, computed as the Fuzzy Reordering Score, implementation obtained from Voita et al. (2021)
    - Token overlap: how many tokens from the source also occur in the target
    - Word overlap: how many words from the source also occur in the target, excluding punctuation

**Additional feature visualisations** We provide the correlations between these features and our memorisation metrics (see Figure 13a), along with the correlations between features (see Figure 13b), and also include additional feature visualisations in Figure 14.

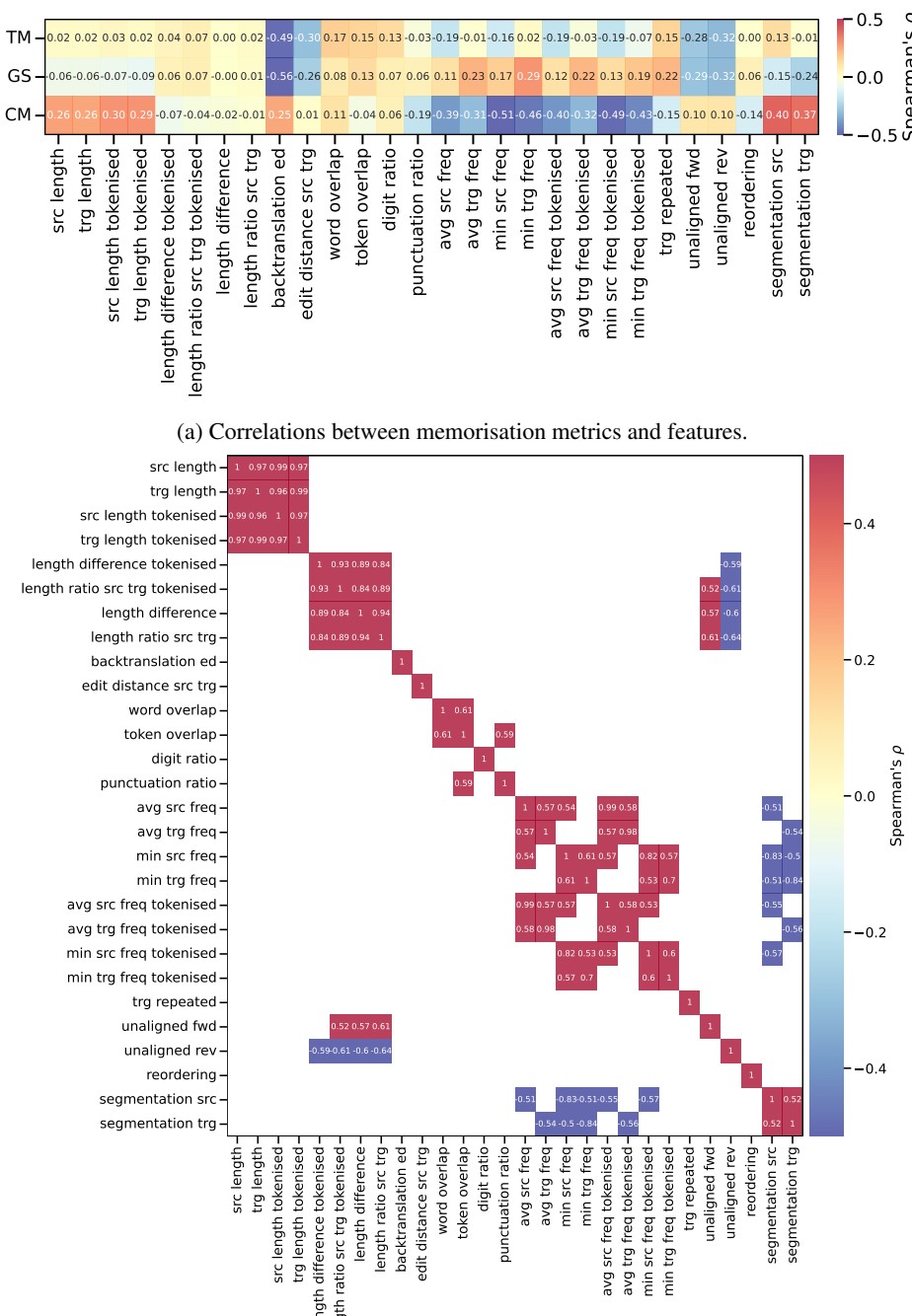

(a) Correlations between memorisation metrics and features.

(b) Correlations among features, displaying correlations >0.5 or <-0.5. The figure is mirrored in the diagonal, but both sides are shown to ease inspection by the viewer.

Figure 13: Correlations between memorisation metrics and features, and correlations among features.

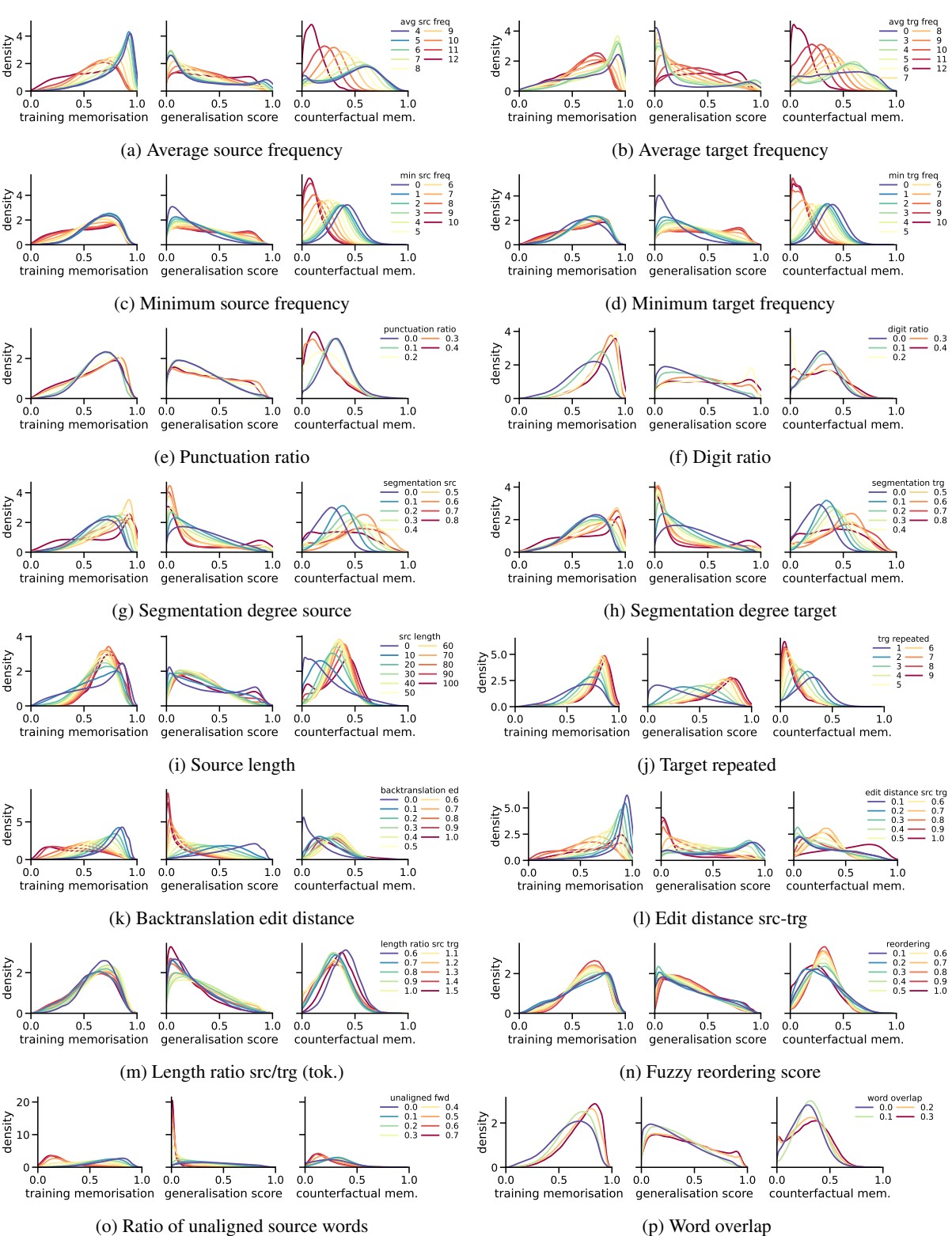

Figure 14: Visualisation of various features. We show their distribution over the three memorisation metrics employed.

**Manual annotation**    We uniformly sample 250 examples from the EN-NL memorisation map, with source lengths $l$ such that $10 < l < 15$. We annotate them using the following labels, where multiple labels can apply to the same example:

- **(Almost) Word for word**: if the target is almost a word-for-word translation of the source, with very minor rephrasing or change in word order. For example "In only days, without food or water, Society collapses into chaos." ↔ "In slechts enkele dagen, zonder eten of drinken, stort de maatschappij in chaos."
- **Paraphrase**: if the target generally expresses the same meaning as the source, but using different wording, e.g. "Let me now make two observations concerning the Green Paper on sea ports." ↔ "Tot slot wil ik nog enkele opmerkingen maken over het Groenboek over havens".
- **Inaccurate**: if the target is an incorrect translation or discusses something that the source does not warrant, e.g. 'sancties' in this source-target pair "We ask you to form a worldwide front against war and NATO." ↔ "Wij vragen u om een wereldwijd front tegen de oorlog en sancties te vormen."
- **Adds content**: if the target introduces new information. "And My curse will be upon you until the Day of Judgment." ↔ "En voorwaar, op jou rust Mijn vervloeking, tot de Dag des Oordeels." Depending on how relevant the added content is, the pair can still be word-for-word / paraphrase / inaccurate.
- **Removes content**: if the target removes content from the source, as is the case in "He married his beloved wife, Penny, in 1977 and raised a family." ↔ "In 1977 trouwde hij met Penny en samen brachten ze een gezin groot." Here 'his beloved wife' is removed in the target. Depending on how relevant the removed content is, the pair can still be word-for-word / paraphrase / inaccurate.
- **Different formatting**: if the target changes the punctuation or the capitalisation, e.g. "It is difficult to negotiate with people who CONFUSE AUSTRIA WITH AUSTRALIA."↔"Samenwerken met mensen die Oostenrijk verwarren met Australië is lastig." This can co-occur with other changes.

To create the visualisation in Figure 4, we post-process the labels in the following way: (1) we create a separate label 'word for word'. This contains the examples that are only annotated with the 'almost word for word' label and no others to filter the examples that are literally word for word, from the ones that do have slight changes, such as formatting changes or the removal of a few words. (2) we restrict 'addition' and 'removal' of content to cases that are not labelled as 'almost word for word', since in those cases the addition/removal is considered very minor and non-essential. This helps us to identify the cases where addition/removal actually affects the meaning difference between the source and target.

## D Additional results

### D.1 Noisy memorisation continuum

In §3 and Appendix A.1 we detailed how we obtained the parallel OPUS data, that has targets for the same source sequences in five different languages, and filters noisy data in the process, such as data that has a lot of overlap between the source and target, data with an extreme length difference between source and target, etc. What happens if we compute memorisation measures on a random OPUS subset, instead? We take the OPUS-100 subset for EN-NL that Zhang et al. (2020) released, containing 1M randomly sampled examples.

**Differences**  Figures 15a and b show the memorisation maps for parallel OPUS and OPUS-100, respectively. The two most striking differences are: (1) OPUS-100 has many more datapoints with a counterfactual memorisation score close to 1 (dark red, in the bottom right corner). (2) There are many more examples with a low generalisation score. This might appear unintuitive to the reader, but it does not necessarily mean the OPUS-100 models are worse in terms of their translations' quality; it just means the dataset is more heterogeneous, and there are more source-target pairs with unexpected tokens in the target (remember that we are computing a geometric mean over the target tokens' probabilities).

These two sets of 1M examples actually have some examples in common. In Figure 15c, we illustrate how the scores compare between parallel OPUS and OPUS-100. These scores do strongly positively correlate, but are still quite different in terms of absolute numbers. Hence, how many examples will be memorised, and what exact score is assigned to an individual example does depend on dataset composition.

**Similarities**  Yet, what is perhaps more relevant is that when we measure the correlations between the features we assigned to datapoints and the memorisation metrics – as is illustrated in Figure 16 – the same patterns emerge as we pointed out in §4, with stronger correlations across the board: (backtranslation) edit distance and unaligned words matter for training memorisation and generalisation score, and length, word frequency and segmentation features matter most for counterfactual memorisation.

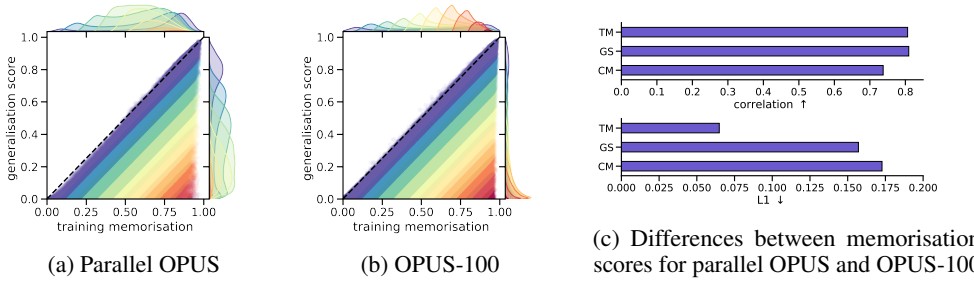

(a) Parallel OPUS            (b) OPUS-100            (c) Differences between memorisation scores for parallel OPUS and OPUS-100

Figure 15: Illustrations describing the differences between the memorisation spectrum for EN-NL, computed using the filtered parallel OPUS data vs. the OPUS-100 data (random OPUS subset).

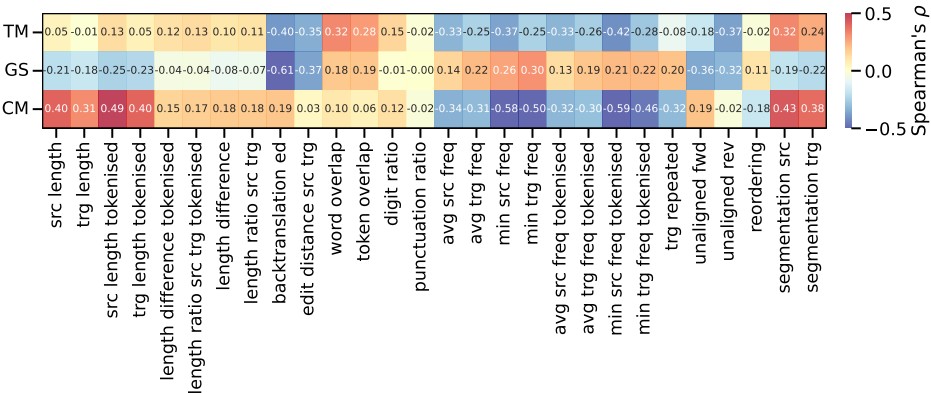

Figure 16: Correlations between memorisation metrics and features.

## D.2 Performance impact

In §6 we observed that examples with higher CM scores tend to be more beneficial for multiple MT performance metrics. In general, this fits with the long tail theory of Feldman (2020); Feldman and Zhang (2020) but why might this be the case? In computer vision classification tasks, Feldman and Zhang (2020) observe that examples with high CM scores are beneficial when making predictions for visually similar test examples. We would like to highlight two additional patterns observed in relation to high CM scores.

Firstly, we would like to examine the log-probability performance impact more closely. Figure 18 provides the log-probability per coordinate, where darker means more relevant. Why are the bottom rows, and the examples with high CM in particular, most relevant? This metric is computed using the *target* tokens' probabilities, that are easily negatively affected if there are some unexpected target tokens. Coordinates in the bottom right might be relevant because they include somewhat 'noisy' data, which increases uncertainty in the model during training, which thus smooths the output probability distribution. To examine whether our data reflects that, we put tokens from the FLORES 'dev' set in buckets based on the mean token probability that they have in the predictions of all models trained in §6.1. We compare these token probabilities to those from models that leave out examples with a certain CM. If examples with high CM are removed during model training (e.g. row 0.7 in Figure 18), the token probabilities for buckets with a relatively low probability decrease. Vice versa, when removing examples with low CM, the token probabilities for buckets with a relatively low probability *increase*. This suggests that by removing examples with a high CM, the output distribution becomes less smooth.

Secondly, we would like to point out that examples with a high CM score generally have less redundancy than examples with a low CM score (in particular, compared to examples with a high training memorisation *and* generalisation score). The corpus that we constructed has 1M *unique* source sentences, so none of them are repeated, but, nonetheless, there are sentences that are more alike than others in terms of $n$-gram count, explaining that redundancy. To illustrate that, Figure 19 conveys the ratio of unique trigrams vs. all trigrams in the data from a particular coordinate.

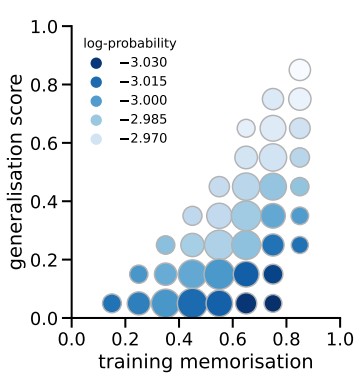

Figure 17: Relevance of regions for the log-probability metric. Darker means more relevant.

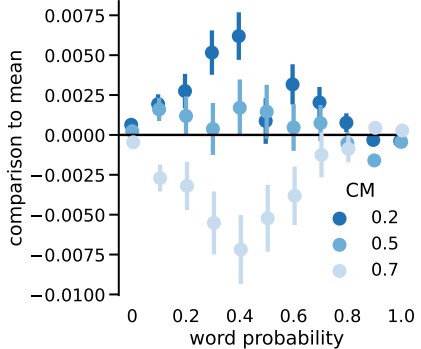

Figure 18: Differences between coordinates with three different CM scores, compared to the mean of all models, per group of words with a certain probability.

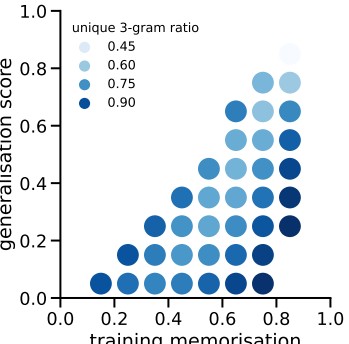

Figure 19: Ratio of unique trigrams over all trigrams, per coordinate.

## D.3 Replacing likelihood with BLEU

Counterfactual memorisation is the difference between the train and test probability of the target. Yet, in NMT, we generate a sequence, and we typically do not use the probability of the target as an adequate measure of success since that metric is severely affected by the sequence length. In the main paper, we instead used the geometric mean of the target token probabilities to compute the memorisation metrics. Here, we consider *generating* sequences using greedy decoding and replace the train and test probabilities with BLEU scores.

Figures 20a and 20b illustrate how the memorisation map changes when we switch to BLEU-based metrics, using the EN-NL data. The examples generally lie closer to the diagonal, and the computation of the memorisation metrics is less stable across models: comparing CM scores from models with 20 seeds to those of 20 other seeds leads to Pearson's $r=0.84$ (it was 0.94 for the LL-based scores). When comparing the two sets of LL- and BLEU-based memorisation metrics, the TM and GS metrics correlate strongly with $\rho = 0.89$, $\rho = 0.80$, although the CM's correlation is substantially lower ($\rho = 0.54$). Examples that the model fully memorises (BLEU>99 or LL>0.9) do reside in the same area on the two maps, as shown by Figures 20c and 20d.

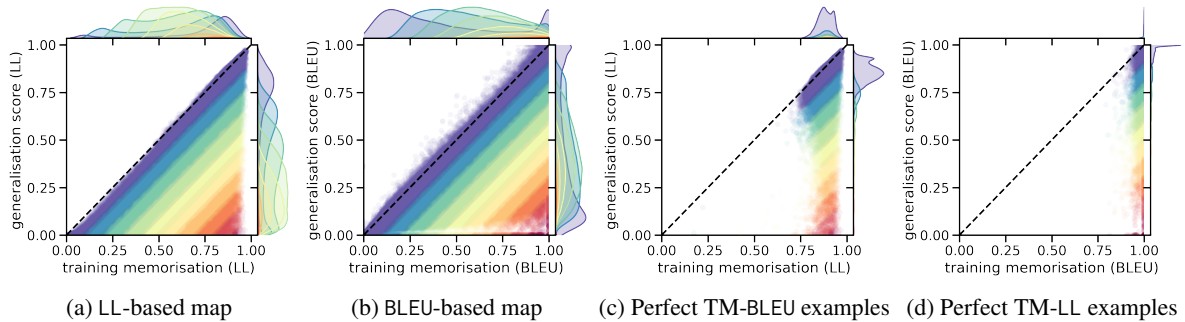

(a) LL-based map  (b) BLEU-based map  (c) Perfect TM-BLEU examples  (d) Perfect TM-LL examples

Figure 20: Illustration of how the memorisation map changes when we compute the memorisation metrics using BLEU instead of LL as a performance metric, for EN-NL. Colour represents CM scores.

In §4, we discussed how various surface-level features of the source and target correlate with the different memorisation metrics. Figure 21 provides the same results, but then for the BLEU-based metrics. Although the majority of the correlations are lower in terms of absolute $\rho$, the same patterns apply in terms of the previously identified positive/negative correlations.

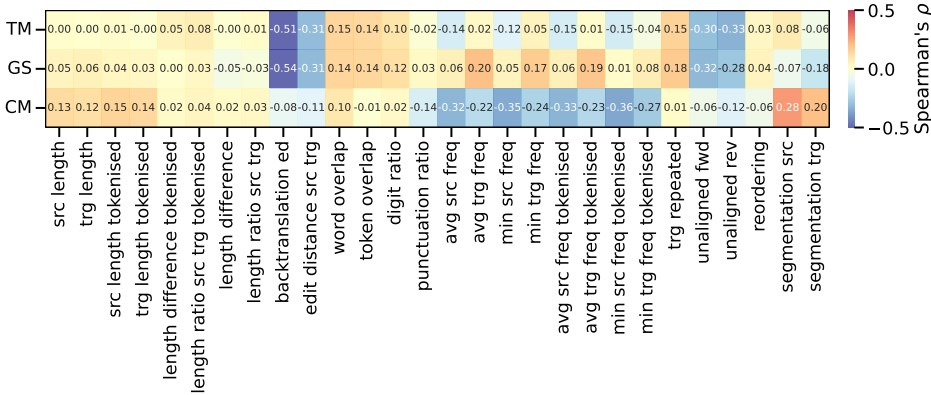

Figure 21: Feature correlations for the BLEU-based memorisation metrics.