# OpenReview forum: "Memorisation Cartography: Mapping out the Memorisation-Generalisation Continuum in Neural Machine Translation"
_EMNLP/2023/Conference — EMNLP 2023 Main_

### Official Review · Reviewer_5nce · 2023-08-04

**Soundness:** 4

**Excitement:**

4: Strong: This paper deepens the understanding of some phenomenon or lowers the barriers to an existing research direction.

**Paper Topic And Main Contributions:**

This paper analyses neural machine translation models by mapping source/target pairs onto a memorisation/generalisation map. To create this map, the authors train multiple systems on overlapping subsets of the data, and then compare the length-normalized probabilities P(Y|X), averaged on subsets where (X,Y) is part of the training data (training memorization (TM)) or not (generalisation score (GS)). The different of these scores is known as the counterfactual memorisation (CM).

The authors identify correlations between these scores and surface-level features of the input pairs, and then build models to predict TM/GS/CM. They also analyze how example from different regions of the map impact performance.

**Questions For The Authors:**

A. You present the correlation across seeds when using half the data. How would the CM scores change if you included larger subsets of the data for training (as overall model performance would likely improve)?

B. In figure 6b, why is it harder to predict CM than TM or GS?

**Reasons To Accept:**

This paper could help researchers better understand the behavior of neural machine translation systems.

The approach is well-motivated and simple. The analysis is quite thorough and presented logically.

This paper could potentially lead to data up/downsampling techniques to improve machine translation quality (although this is not yet established).

**Reasons To Reject:**

The findings would be more convincing if there were models with English as the target language (i.e. opposite direction).

The datasets are quite small, although this is understandable given the high compute requirements. The relationship between TM/GS vs model/dataset sizes is unclear.

**Reproducibility:**

3: Could reproduce the results with some difficulty. The settings of parameters are underspecified or subjectively determined; the training/evaluation data are not widely available.

**Reviewer Confidence:**

4: Quite sure. I tried to check the important points carefully. It's unlikely, though conceivable, that I missed something that should affect my ratings.

**Typos Grammar Style And Presentation Improvements:**

The curve labels in figure 3 should be ranges of values, not a single scalar. If that would make the figure too cluttered, maybe expand the description of the figure.

---

> ### Author Rebuttal · Authors · 2023-08-29
>
> Great to hear that you think our work is thorough and well-motivated. We hope to contribute to a better understanding of NMT systems and encourage more research into memorisation for other NLP tasks, as well.
>
> During the discussion period, we hope that you could elaborate on your curiosity about English on the target side. Our own expectation is that that would lead to results that are quantitatively different but qualitatively largely the same.
>
> **Regarding question A:** When computing CM scores using models trained with larger training sets, the main difference we expect to see in the CM scores is that data points receive scores that are closer together. As you point out, the overall model performance would improve (and thus the GS score would improve), but if the model capacity doesn’t change, the TM score might slightly decrease, thus bringing the CM scores closer together across the board.
>
> **Regarding question B:** CM is a combination of TM and GS (where the latter is subtracted from the former). If the model inaccurately predicts TM and inaccurately predicts GS, those errors compound when the model predicts CM. As a result, it is not unexpected that that metric is harder to predict using an MLP.
>
> Thanks for your suggestion regarding improving Figure 3, we can definitely implement that!

---

### Official Review · Reviewer_Jciq · 2023-08-08

**Soundness:** 4

**Excitement:**

4: Strong: This paper deepens the understanding of some phenomenon or lowers the barriers to an existing research direction.

**Paper Topic And Main Contributions:**

This paper addresses two questions for neural machine translation models, which are what determines a datapoint’s position on that spectrum, and how does that spectrum influence neural models’ performance. The experimental results illustrate that memorization is not one-dimensional.

**Reasons To Accept:**

(1) Overall, the paper is written well and easy to follow; (2) It shows detailed experimental results and discussions about memorization in Neural Machine Translation; (3) The experiments are sufficient and the results are comprehensive and convincing.

**Reasons To Reject:**

The novelty is limited and the motivation is not strong.

**Reproducibility:**

4: Could mostly reproduce the results, but there may be some variation because of sample variance or minor variations in their interpretation of the protocol or method.

**Reviewer Confidence:**

4: Quite sure. I tried to check the important points carefully. It's unlikely, though conceivable, that I missed something that should affect my ratings.

---

> ### Author Rebuttal · Authors · 2023-08-28
>
> Thanks for the acknowledgement of our detailed experimentation; we hope our results can contribute in a positive way to memorisation studies in general but can also benefit machine translation and sequence-to-sequence learning communities.
>
> To reiterate our novelty and motivation from a fresh perspective:
> - Our main **motivation** is the need to understand memorisation – what leads a model to memorise some data points, without ever learning some others – and the need to understand whether or not this memorisation is problematic, i.e. whether it negatively influences the model’s performance.
> - The **novelty** lies in approaching memorisation as a graded phenomenon, examining datapoints characteristics across the entire training dataset, instead of focusing on a small percentage of the data that has been marked as problematic, as is the predominant approach in the related work (see Section 2 of our paper).

---

### Official Review · Reviewer_pPXn · 2023-08-11

**Soundness:** 4

**Excitement:**

3: Ambivalent: It has merits (e.g., it reports state-of-the-art results, the idea is nice), but there are key weaknesses (e.g., it describes incremental work), and it can significantly benefit from another round of revision. However, I won't object to accepting it if my co-reviewers champion it.

**Paper Topic And Main Contributions:**

This paper delves into the intricate dynamics of memorization and generalization in the realm of Neural Machine Translation (NMT). By mapping out the continuum between these two facets, the authors aim to unravel the nuanced trade-offs that models make when faced with the dual challenges of retaining specific training examples and generalizing to unseen data. It also presents a meticulous exploration of the intricacies of NMT models, shedding light on their behaviors and propensities. Such insights are invaluable in advancing the field of machine translation and in optimizing model performance by striking the right balance between memorization and generalization.

**Questions For The Authors:**

Question A: The paper introduces the "Memorisation Cartography" approach to understand the continuum between memorization and generalization. How do you envision this methodology being adapted or extended to other NLP tasks beyond NMT?
Question B: The experiments presented, while comprehensive, primarily focus on certain datasets and languages. Could you elaborate on how the findings might generalize to low-resource languages or more diverse domains?
Question C: Given the deep dive into the memorization-generalization continuum, are there other factors or considerations in NMT that might interact with these phenomena? How might these interactions influence the overall performance and behavior of NMT systems?

**Reasons To Accept:**

1. Innovative Exploration of Memorization vs. Generalization: The paper offers a groundbreaking perspective on the memorization-generalization continuum in NMT, supported by the novel "Memorisation Cartography" approach. This deep dive into a fundamental dilemma is both timely and essential for the evolving NLP landscape.

2. Robust Experimental Foundation with Broad Implications: The thorough experiments presented validate the research's claims and offer insights with wide-ranging applications. These findings can guide the design of efficient NMT architectures, especially beneficial in low-resource scenarios, and pave the way for optimized model performance.

3. Significant Theoretical and Practical Contributions: Beyond its immediate application to NMT, this research enriches the theoretical understanding of neural network behaviors. It also opens avenues for future work, with potential extensions to various NLP tasks and strategies

**Reasons To Reject:**

1. Potential Lack of Generality: While the paper offers insights into the memorization-generalization continuum in NMT, it might be constrained in its applicability. The findings, although significant, might not extend seamlessly to other NLP tasks or even to all NMT architectures.

2. Need for Further Validation: Although the paper presents various experiments, additional validation on a broader range of datasets, including more diverse languages and domains, would have strengthened the claims. The current experimental setup might not capture all nuances and complexities of real-world translation tasks.

3. Absence of Practical Implementations: The paper, while rich in theoretical content, lacks a clear bridge to practical implementation. Concrete strategies or tools that harness the presented insights for real-world applications would have elevated the paper's utility.

4. Depth over Breadth: The paper's deep dive into the memorization-generalization continuum, while commendable, might come at the expense of exploring other equally pertinent aspects of NMT. A more holistic approach, considering other factors influencing NMT performance, could have provided a rounded perspective.

**Reproducibility:**

3: Could reproduce the results with some difficulty. The settings of parameters are underspecified or subjectively determined; the training/evaluation data are not widely available.

**Reviewer Confidence:**

3: Pretty sure, but there's a chance I missed something. Although I have a good feel for this area in general, I did not carefully check the paper's details, e.g., the math, experimental design, or novelty.

---

> ### Author Rebuttal · Authors · 2023-08-28
>
> Many thanks for the elaborate review; your comments encourage us to elaborate on the wider applicability of the experiments and our approach in the paper.
>
> **Regarding the depth of the article, the need for further validation and lack of generality**: We opted for an in-depth analysis of the presented memorisation maps without experimenting with training procedures and model architectures. Yet, we do present experiments for 5 languages (main paper), and also included further experimentation with changing the data (Appendix D.1) and changing the way memorisation is measured (Appendix D.3) for a subset of the experiments, which confirmed that our findings are qualitatively robust. In fact, our training data selection procedure (creating parallel corpora) was tailored to validate that the results hold within this group of languages, and the training data comes from a variety of domains (see Question A). Those parallel corpora could not be constructed for low-resource languages. While we recommend further experimentation with different languages and larger corpora, the training procedure and model architecture represent standard practices in the field. As a result, we maintain that our findings provide appealing lessons for NMT practitioners as is. Note also that our experimentation required 1.6k GPU hours, which is why we prioritised experimenting with languages and data over evaluation across architectures, which might not have been the best use of our resources.
>
> **Regarding the absence of practical implementations:** The main focus of our work is, indeed, analysing the memorisation maps. We do provide a small proof-of-concept implementation in section 6.2, and would be happy to be more explicit about further suggestions for practical implementations. To name a few thoughts that we have:
> - The training data that is easiest to translate (top right corner) can make a model overconfident in terms of its log-probabilities (section 6.1, appendix D.2). Downsample or filter data with word-for-word translations prior to model training.
> - Source-target pairs with low overlap or high backtranslation distances have low TM scores. Potentially, one could improve NMT system's ability to paraphrase by upweighing examples in the lower left corner of the memorisation map.
> - The memorisation map can be used to create a learning curriculum for NMT. To give an example: data points from the top right corner are taught first (`basic translations’), followed by examples with a high CM score.
>
> **Regarding question A:** The metrics we used to compute the memorisation map were NMT specific, but the approach as a whole could easily be applied to other tasks; one would have to redefine TM and GS, and compute task-specific datapoint features. We would be very interested in seeing more memorisation maps and explorations of which features drive data points’ positions on the maps for a range of NLP tasks. In particular, we recommend explorations that contrast sequence classification and sequence-to-sequence learning and, more generally, comparisons of multiple tasks that use the same data. After all, whether a data point requires memorisation or not depends on the task one is working on. We hope our study can provide an in-depth example of how to set up such a study and will inspire others to ask similar questions for other tasks.
>
> **Regarding question B:** Firstly, the data we used is already quite diverse since OPUS includes data from many different sources (among others, Wikipedia, books, WMT-News, OpenSubtitles, web-scraped text / ParaCrawl, Europarl). The data used was filtered (see Appendix A), and we do examine the effect of using randomly sampled data for a subset of the experiments (see Appendix D.1), in which case our results remain qualitatively similar. Our results thus do generalise when varying some parameters of the experimental setup.
>
> Still, when making substantial changes to the experimental setup, such as using very low-resource languages, the findings might differ. The memorisation scores obtained depend on the **ratio** of the dataset size and the model capacity. For lower resource languages, the model will be able to obtain higher training memorisation scores across the board, and the examples from the lower left corner of the memorisation map will move to the right. This will influence the results obtained. Examples with a high CM score may no longer be beneficial in terms of model performance when that ratio becomes too small, but further studies would be required to confirm that hypothesis.
>
> **Regarding question C:** Our approach in the paper is data-centric; we examine the role of data points in the memorisation process, but, of course, there are other factors contributing to memorisation. An obvious consideration is training time: two systems with similar validation BLEU scores can be very different in terms of memorisation (maps). Other considerations concern the computation of the loss (label smoothing explicitly discourages memorisation, whereas teacher forcing might promote memorisation) and the type of model used. We do not experiment with the training procedure in the paper but encourage future work to take a model-centric approach to memorisation since the data and model interact when enabling memorisation.
>
> **Regarding the reproducibility**: We noticed that you indicated that the results might be hard to reproduce. We would like to point out that we included the anonymised codebase in the paper, discussed GPU architecture details and training commands in Appendix A, and that the training data from OPUS is publicly available. Due to size restrictions, we cannot put the memorisation map in the codebase, but we aim to make the numbers we obtained available to interested readers for further analysis, and already provide a subset of the memorisation maps through the public demo. Could you indicate what in the paper we could improve to make our work more easily reproducible?

---

### Official Review · Reviewer_cDFT · 2023-08-11

**Soundness:** 4

**Excitement:**

4: Strong: This paper deepens the understanding of some phenomenon or lowers the barriers to an existing research direction.

**Paper Topic And Main Contributions:**

This work builds a memorisation map and investigates the relation among datapoint characteristics, Counterfactual Memorisation (CM) metric, and model performance in a multilingual neural machine translation scenario by large-scale experiments. The empirical results reveal new observations:
1) the memorisation can be influenced differently by characteristics of a datapoint.
2) the memorisation can be predicted by characteristics of a datapoint.
that can be instructive to researchers and other generation tasks.
and also validate some findings:
1) datapoints with high CM value are more beneficial to model training.
2) examples with high CM scores lead to more hallucinations.

**Questions For The Authors:**

1. Line 501, what does "we create training subsets by removing the nearest examples" mean?
2. Figure 8, why are the areas of log-probability and BLEU/hallucination tendency so different? It seems that the areas of log-probability are more similar to Figure 7(a).

**Reasons To Accept:**

1. The article is well written and structured.

2. The article has sufficient experimental studies to demonstrate the findings.

3. The research question this work explored is useful to understand memorisation behaviour and solve hallucination problem which are widely existed in current LLMs. The work can also help reseachers to improve the model performance by manipulating used datapoints.

**Reasons To Reject:**

1. As the authors stated in the Limitations, though computationally expensive experiments are conducted, the applicability of the findings this work observed is questionable as the experimental setup is limited.

**Reproducibility:**

3: Could reproduce the results with some difficulty. The settings of parameters are underspecified or subjectively determined; the training/evaluation data are not widely available.

**Reviewer Confidence:**

3: Pretty sure, but there's a chance I missed something. Although I have a good feel for this area in general, I did not carefully check the paper's details, e.g., the math, experimental design, or novelty.

---

> ### Author Rebuttal · Authors · 2023-08-28
>
> We’re glad to hear you find the research question worth exploring!
>
> **Regarding the limited applicability of our findings**: we indeed point out that the counterfactual memorisation values we obtain are specific to our experimental setup. However, at the same time, we do demonstrate that the relation between datapoints’ characteristics and high/low training memorisation, generalisation score and counterfactual memorisation values are robust. We show that in three ways:
> 1. by computing across-language generalisation of the MLPs trained to predict TM, GS and CM from datapoints’ characteristics (see Section 5)
> 2. by replacing the training data with randomly selected OPUS data (as opposed to filtered data) in appendix D.1
> 3. by replacing the likelihood metric (see line 187) with BLEU scores in the computation of TM, GS and CM in appendix D.3
> In each of these scenarios, the salience of datapoints’ characteristics is similar, demonstrating robustness against changes in the experimental setup.
>
> **Regarding question 1, line 501**: In section 6.1, we train NMT models on the same training data as used in section 3, but with some examples removed (up to 750k tokens, which is approximately 3-5% of the input-output pairs). Which of the examples are removed depends on the coordinates from the memorisation map: we create such modified training sets for 55 coordinates total (all coordinates in the lower right triangle of the memorisation map) and when making the modified training set for one coordinate, we remove the 750k tokens closest to that coordinate on the memorisation map.
>
> Let me give an example to clarify how this works: To create the training subset for coordinate (0.2, 0.1), we go to the memorisation map, take the examples with a TM value of ~0.2, and a GS value of ~0.1, until we have the 750k tokens closest to coordinate (0.2, 0.1). We remove them from the training data, and train the model. The lower the performance of this system, the more beneficial those 750k tokens apparently were.
>
> **Regarding question 2**: we opted for two different areas because log-probability has, relatively speaking more `most-beneficial’ coordinates in the bottom half and the lower left corner compared to BLEU and the hallucination tendency. Log-probability’s most beneficial areas can also be seen in Figure 17, appendix D.2.

---

### Meta-Review · Area_Chair_E8Lp · 2023-09-19

**Recommendation:** 5

**Metareview:**

There is clear consensus amongst the reviewers that this is a strong paper both in terms of Soundness and Excitement. The weaknesses and questions raised by the reviewers were sufficiently answered by the authors, as is clear in the review scores and discussion post-rebuttal. The remaining weaknesses are mainly suggestions that would make the work even better, rather than fill in some big gap in the current work. Overall the paper is well written, proposed methodology novel and transferable, backed by sufficient experimentation and targets an important problem in the field of NLP. It has _strong soundness_ and _strong excitement_.

The following is a summary of the strengths, weaknesses and scores across the four reviews:

**Strengths:**

- Well written and well structured (**cDFT**, **Jciq**, **5nce**)
- Claims made in the paper are backed by sufficient experimental results (**cDFT**, **pPXn**, **Jciq**, **5nce**)
- Paper works towards an important problem of generalization/memorization/hallucinations, which are very pertinent to the current state of NLP (**cDFT**, **pPXn**)
- Provides actionable suggestions on how one can improve models by modifying their data (**cDFT**, **pPXn**, **5nce**)
- Suggested methodology of memorization mapping is novel and brings a new perspective to knowledge within a network (**pPXn**, **cDFT**, **Jciq**)

**Weaknesses:**

- Application to low resource settings is not obvious (**pPXn**)
- Reviewers agreed that the following are *not major weaknesses* after rebuttal:
	- Experimental design limits the findings to the specific models and settings (**cDFT**, **pPXn**, **5nce**)
		- Rebuttal: Authors agree, but also note (and reviewers allude to as well) that the experimentation targets a few languages and datasets, and goes into depth in each of these instead of covering a breadth of models. However, the depth of experiments already uncovers a new perspective sufficiently.
	- Breadth of the experimentation is limited in lieu of depth (**pPXn**, **5nce**)
		- Rebuttal: The depth of experiments already reveals a lot of insights, and the breadth was inherently limited by computational cost


**Scores in decreasing order of confidence:**

|      | Soundness | Excitement | Reproducibility | Confidence | Additional Notes             |
|------|-----------|------------|-----------------|------------|------------------------------|
| 5nce | 4         | 4          | 3               | 4          |                              |
| Jciq | 4         | 4          | 4               | 4          | Did not acknowledge rebuttal |
| cDFT | 4         | 4          | 3               | 3          |                              |
| pPXn | 4         | 3          | 3               | 3          |                              |

---

### Decision · Program_Chairs · 2023-10-07

**Decision:**

Accept-Main

**Comment:**

There is clear consensus amongst the reviewers that this is a strong paper both in terms of Soundness and Excitement. The weaknesses and questions raised by the reviewers were sufficiently answered by the authors, as is clear in the review scores and discussion post-rebuttal. The remaining weaknesses are mainly suggestions that would make the work even better, rather than fill in some big gap in the current work. Overall the paper is well written, proposed methodology novel and transferable, backed by sufficient experimentation and targets an important problem in the field of NLP. It has _strong soundness_ and _strong excitement_.

The following is a summary of the strengths, weaknesses and scores across the four reviews:

**Strengths:**

- Well written and well structured (**cDFT**, **Jciq**, **5nce**)
- Claims made in the paper are backed by sufficient experimental results (**cDFT**, **pPXn**, **Jciq**, **5nce**)
- Paper works towards an important problem of generalization/memorization/hallucinations, which are very pertinent to the current state of NLP (**cDFT**, **pPXn**)
- Provides actionable suggestions on how one can improve models by modifying their data (**cDFT**, **pPXn**, **5nce**)
- Suggested methodology of memorization mapping is novel and brings a new perspective to knowledge within a network (**pPXn**, **cDFT**, **Jciq**)

**Weaknesses:**

- Application to low resource settings is not obvious (**pPXn**)
- Reviewers agreed that the following are *not major weaknesses* after rebuttal:
	- Experimental design limits the findings to the specific models and settings (**cDFT**, **pPXn**, **5nce**)
		- Rebuttal: Authors agree, but also note (and reviewers allude to as well) that the experimentation targets a few languages and datasets, and goes into depth in each of these instead of covering a breadth of models. However, the depth of experiments already uncovers a new perspective sufficiently.
	- Breadth of the experimentation is limited in lieu of depth (**pPXn**, **5nce**)
		- Rebuttal: The depth of experiments already reveals a lot of insights, and the breadth was inherently limited by computational cost


**Scores in decreasing order of confidence:**

|      | Soundness | Excitement | Reproducibility | Confidence | Additional Notes             |
|------|-----------|------------|-----------------|------------|------------------------------|
| 5nce | 4         | 4          | 3               | 4          |                              |
| Jciq | 4         | 4          | 4               | 4          | Did not acknowledge rebuttal |
| cDFT | 4         | 4          | 3               | 3          |                              |
| pPXn | 4         | 3          | 3               | 3          |                              |